# A Comparative Study about Vertical Accuracy of Four Freely Available Digital Elevation Models: A Case Study in the Balsas River Watershed, Brazil †

**Zuleide Alves Ferreira** [1,2,*] **and Pedro Cabral** [1]

1   NOVA Information Management School (NOVA IMS), Universidade Nova de Lisboa, Campus de Campolide, 1070-312 Lisbon, Portugal; pcabral@novaims.unl.pt
2   Insituto Federal de Educação, Ciência e Tecnologia do Tocantins, Campus Palmas, Palmas 77021-090, Brazil
*   Correspondence: zuleide@ifto.edu.br
†   Presented at 7th International Conference on Geographical Information Systems Theory, Applications and Management, Online, 23–25 April 2021. Available online https://gistam.scitevents.org/Home.aspx?y=2021 (accessed on 29 December 2021).

**Abstract:** Digital elevation models (DEMs) provide important support to research since these data are freely available for almost all areas of the terrestrial surface. Thus, it is important to assess their accuracy for correct applicability regarding the correct use scale. Therefore, this paper aims to assess the vertical accuracy of ALOS PALSAR, GMTED2010, SRTM, and Topodata DEMs according to the Brazilian Cartographic Accuracy Standard through the official high accuracy network data of the Brazilian Geodetic System. This study also seeks to investigate whether the altimetric error is correlated with altitude and slope in the study area. Our results showed that the four assessed DEMs in this study demonstrated satisfactory accuracy to provide mappings in scales up to 1:100,000 because more than 90% of the extracted points presented altimetric errors of less than 25 m when compared with the reference points from the high accuracy network of the Brazilian Geodetic System. Regarding the altimetric error, we could not find a significant correlation with altitude or slope in the study area. In this sense, future DEMs assessments should be based on the investigation of other factors that may influence altimetric error.

**Keywords:** DEM; assessment; altitude; ALOS PALSAR; GMTED2010; SRTM; Topodata

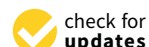

## 1. Introduction

Digital elevation models (DEMs) provide an important topographic product that is fundamental for many scientific and commercial applications [1,2]. However, traditional methods to acquire information for DEM generation are often expensive and time-consuming due to land surveying necessity [2]. On the other hand, several DEM products from many sources have been made freely available to geoinformation users in the last decade, so it is important to investigate their possible applications by assessing their accuracy [3].

DEM products accuracy has been regularly investigated to evaluate their applicative potentialities, thus improving mapping methods [4]. Most of these experiments are performed by comparing the extracted data from DEMs to a set of reference data, i.e., control points, through accuracy statistical indicators, such as mean difference, standard deviation, or root mean square error [4].

DEMs accuracy assessment requires further attention considering that, despite technological advances in the creation and availability of these products, there are still no specific standardized guidelines regarding this assessment process [5]. Nonetheless, in Brazil, there is a decree that regulates the quality of cartographic products by establishing instructions for the technical standards of national cartography. The Decree n° 89,817/1984 determines criteria for cartographic products classification regarding their accuracy and the distribution of errors using a statistical indicator of positional quality named the "Cartographic

Accuracy Standard" (*Padrão de Exatidão Cartográfica—PEC*). Therefore, 90% of the extracted points from the cartographic product must not present errors higher than those predicted in the PEC when their coordinates are compared with those from surveyed points in the field through a high accuracy method [6,7].

There are a lot of studies about DEMs accuracy assessment [8–12]. However, none have assessed the vertical accuracy of the ALOS PALSAR, GMTED2010, SRTM, and Topodata DEMs according to the Brazilian Cartographic Accuracy Standard (PEC). Thus, the purpose of this study is to assess the vertical accuracy of the above-mentioned DEMs by using the official high accuracy network data of the Brazilian Geodetic System. This study also seeks to investigate whether the altimetric error is correlated with altitude and slope in the study area. We expect that results contribute to the correct applicability of the analyzed DEMs according to an appropriate use scale in Brazil and other places dealing with the same problem context.

## 2. Materials and Methods

### 2.1. Study Area

The Balsas River watershed covers 13 municipalities, and its area is 12,352.50 km$^2$, corresponding to nearly 4.5% of the State of Tocantins (Figure 1) [13]. Its altitudes are approximately between 200 and 800 m considering sea level, and inside this area, we can find 105 stations of the official Brazilian geodetic network situated along the main highways of the region (Figure 2). It is worth noting the absence of high accuracy three-dimensional data available for free to the community in various regions of the planet. In this sense, the Balsas River watershed was selected due to the lack of accurate three-dimensional data available for this area.

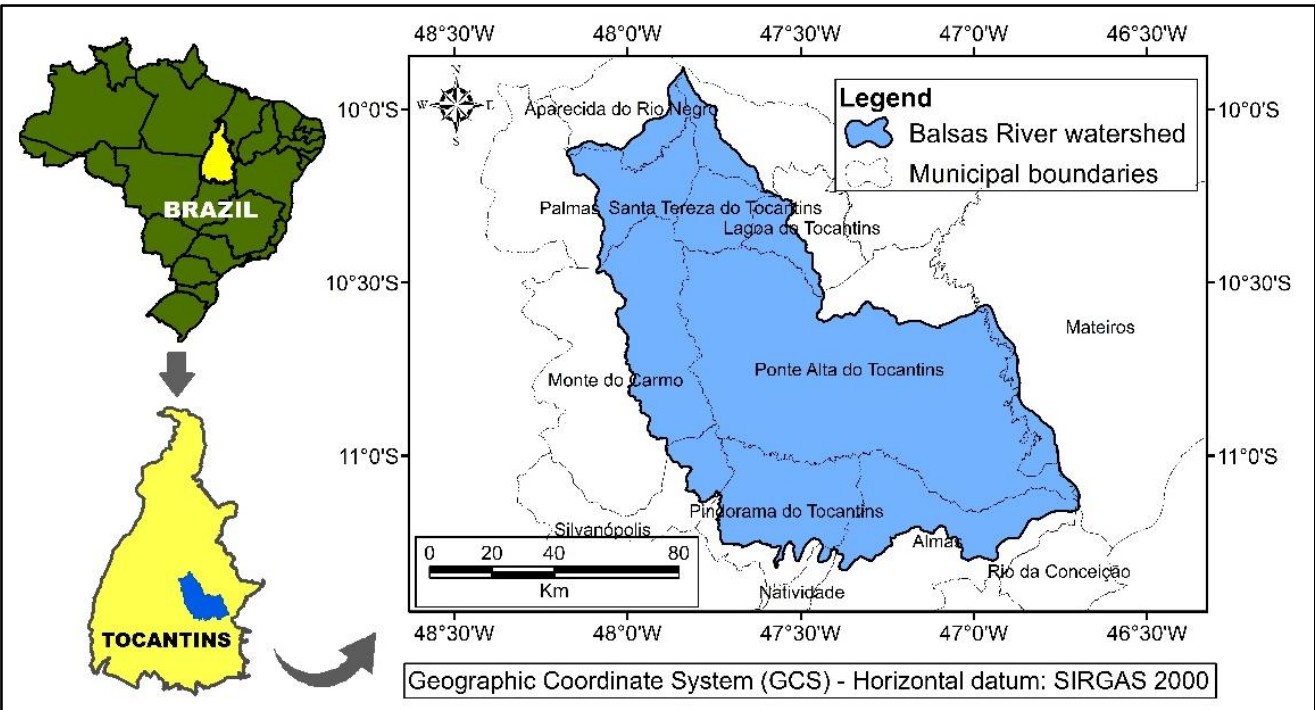

**Figure 1.** Study area.

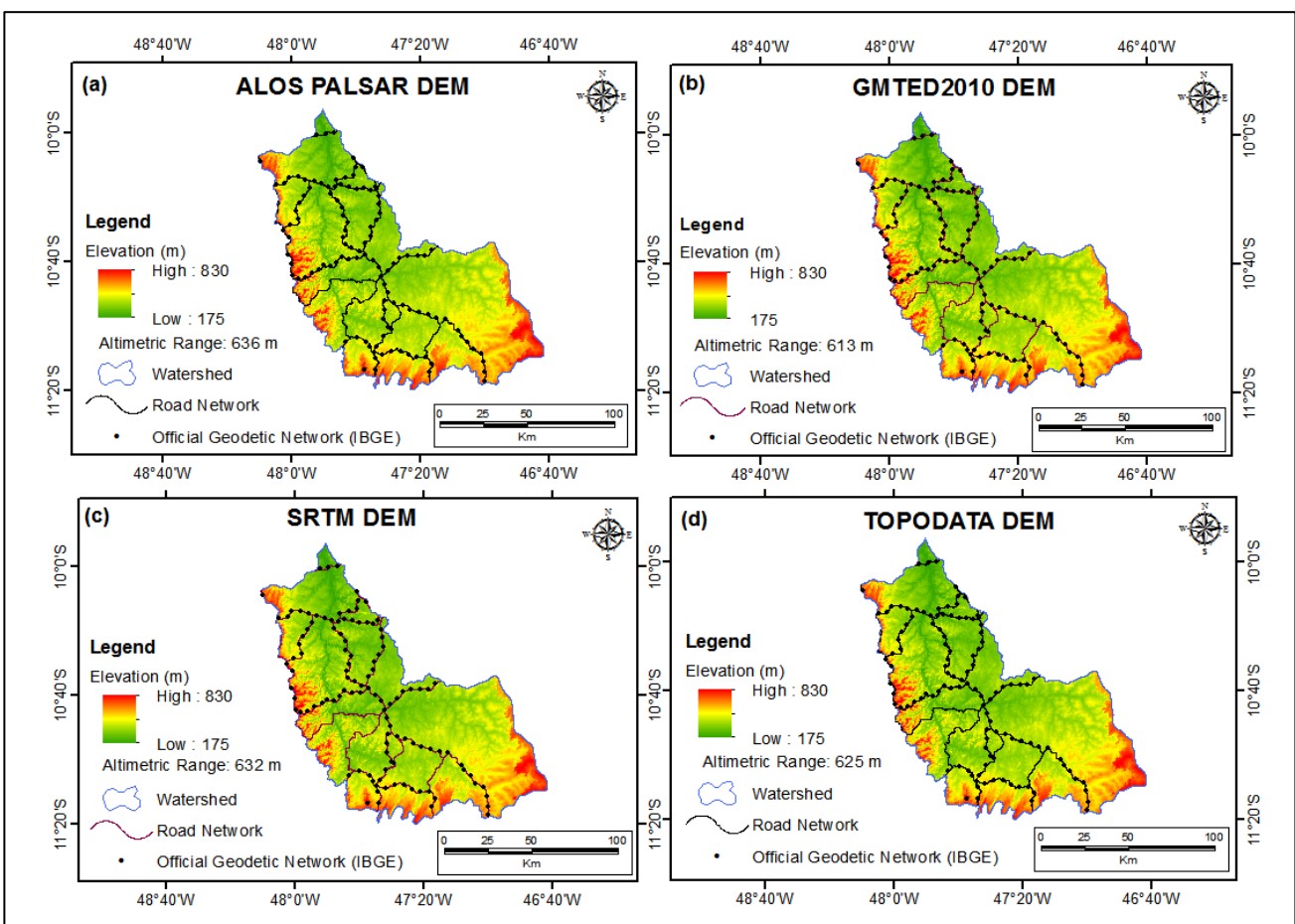

**Figure 2.** Hypsometric maps of Balsas River watershed derived from: (**a**) ALOS PALSAR, (**b**) GMTED2010, (**c**) SRTM, and (**d**) Topodata.

### 2.2. Data

In this accuracy assessment, we compared the extracted points from the four DEMs with the official network data of the Brazilian Geodetic System. This network is composed of geodesic stations located along the main highways throughout the Brazilian territory, which was implemented in 1945 through the high accuracy geometric leveling method [14]. Since then, these altitudes have been regularly recalculated owing to the addition of new geometric leveling lines and the development of new data measurement and processing techniques, in which new observations of geometric leveling and gravimetry are added aiming to ensure the integrity, consistency, and reliability of the information from the Geodetic Database. According to the quality assessment of these altimetric data performed in 2018, 87.5% of the adjusted geopotential values presented standard deviations between 6 and 10 cm in absolute terms [14].

The Advanced Land Observing Satellite "DAICHI" (ALOS) was designed to supply land coverage mapping, resource surveying, and disaster monitoring [15]. It was launched on 24 January 2006, from the Tanegashima Space Center with three sensors onboard, namely, the Panchromatic Remote-Sensing Instrument for Stereo Mapping (PRISM), Advanced Visible and Near Infrared Radiometer type 2 (AVNIR-2), and Phased Array type L-band Synthetic Aperture Radar (PALSAR). The ALOS mission was completed on 12 May 2011, but during its 5-year operation, it shot 6.5 million scenes around the Earth, which have been used in many fields, such as agriculture, natural environment maintenance, forest monitoring, and disaster mitigation [15]. The PRISM sensor is a panchromatic radiometer and has three sets of optical systems with 2.5 m spatial resolution at nadir; the AVNIR-2 sensor is a visible and near-infrared radiometer that provides 10 m spatial resolution

images; and PALSAR is an active microwave sensor that uses L-band frequency to obtain cloud-free and day-and-night land observation [15,16].

The acquired data during the ALOS mission were geometrically and radiometrically corrected. Firstly, the geometric distortions were corrected using some DEMs, and then the radiometry correction was executed by adjusting the brightness of the individual SAR image pixels in the affected foreshortening and layover regions [17,18]. Succeeding the radiometric terrain correction, these products were distributed at two resolutions, 12.5 and 30 m pixel size, generated from high-resolution (NED13) and mid-resolution DEMs (SRTM30, NED1, and NED2), respectively [18].

The Shuttle Radar Topography Mission (SRTM) is an international project developed by the National Aeronautics and Space Administration (NASA) and the National Geospatial-Intelligence Agency (NGA). This mission started on 11 February 2000, and during 10 days, SRTM acquired data over approximately 80 percent of the Earth's land surface through two radar antennas to create the first near-global data set of land elevations [19]. Initially, SRTM data were made publicly available at 3 arc-seconds resolution or 90 m of pixel size for regions outside the United States. However, in 2014, the topographic data were released globally with the full resolution originally measured, that is, 1 arc-second (30 m) [19].

The Topodata project is a topographic database generated from the refinement of SRTM data. Due to the general lack of topographic data at adequate scales in some Brazilian regions, this project was released in 2008 aiming to refine SRTM data from the 3 arc-seconds to 1 arc-second resolution through kriging techniques as well as to provide the derivation of geomorphometric data for the whole Brazilian territory [20,21]. The Topodata project resulted in an extensive structured database freely available for the scientific community, which offers several products such as slope, slopes orientation, horizontal curvature, vertical curvature, and inputs for the drainage structure design, among others [21].

The Global Multi-Resolution Terrain Elevation Data 2010 (GMTED2010) was developed by the United States Geological Survey (USGS) in partnership with the National Geospatial-Intelligence Agency (NGA) to replace the Global 30 Arc-Second Elevation (GTOPO30) as the elevation dataset for global and continental scale applications [22]. GMTED2010 was elaborated using derived data from 11 raster-based elevation sources (Table 1), which provides global coverage from latitude 84° N to 56° S for most products at three different resolutions, 7.5, 15, and 30 arc-seconds, that correspond to nearly 250, 500, and 1000 m of pixel size, respectively [22]. In this study, we selected the GMTED2010 product available in 7.5 arc-seconds resolution, which is widely used in several scientific studies [11,23–29] despite its bigger pixel size when compared with SRTM, for instance. Table 2 presents the original main characteristics of the four DEMs assessed in this study.

**Table 1.** GMTED2010—input source data characteristics (adapted from [22]).

| Dataset | Resolution | Horizontal Unit | Horizontal Datum |
|---|---|---|---|
| SRTM DTED® 2 | 1 | Arc-second | WGS 84 |
| DTED® 1 | 3 | Arc-second | WGS 84 |
| CDED1 | 0.75 | Arc-second | NAD 83 |
| CDED3 | 3 | Arc-second | NAD 83 |
| 15-arc-second SPOT 5 Reference3D | 0.00416666 | Decimal degree | WGS 84 |
| NED | 0.00027777 | Decimal degree | NAD 83 |
| NED—Alaska | 0.00055555 | Decimal degree | NAD 83 |
| GEODATA 9-s DEM version 2 | 0.0025 | Decimal degree | GDA 94 |
| Greenland satellite radar altimeter DEM | 1,000 | Meter | WGS 84 |
| Antarctica satellite radar and laser altimeter DEM | 1,000 | Meter | WGS 84 |
| GTOPO30 | 0.00833333 | Decimal degree | WGS 84 |

DTED®, Digital Terrain Elevation Data; WGS 84, World Geodetic System 1984; CDED, Canadian Digital Elevation Data; NAD 83, North American Datum of 1983; SPOT, Satellite Pour l'Observation de la Terre; NED, National Elevation Dataset; DEM, digital elevation model; GDA 94, Geocentric Datum of Australia 1994; GTOPO30, Global 30-Arc-Second Elevation Dataset.

**Table 2.** Original characteristics of the four assessed DEMs.

| DEM | Coordinate System | Horizontal Datum | Vertical Reference | Pixel Size | Radiometric Resolution |
|---|---|---|---|---|---|
| ALOS PALSAR | UTM | WGS 84 | Ellipsoid * | 12.5 m | 16 bits (signed integer) |
| GMTED2010 | Geographic | WGS 84 | Geoid (EGM96) | 231 m (7.5 arc-seconds) | 16 bits (signed integer) |
| SRTM | Geographic | WGS 84 | Geoid (EGM96) | 30 m (1 arc-second) | 16 bits (signed integer) |
| Topodata | Geographic | WGS 84 | Geoid (EGM96) | 30 m (1 arc-second) | 32 bits (floating point) |

* The orthometric heights with EGM96 vertical datum were converted to ellipsoid heights using the ASF MapReady tool named "geoid_adjust" [17].

### 2.3. Methods

Figure 3 summarizes the methodology used in this study. Firstly, we downloaded the data from the study area, such as raster DEMs and Brazilian official geodetic network points. Then, we proceeded with the radiometric resolution conversion of the Topodata DEM from 32 bits (floating point) to 16 bits (signed integer) to standardize the data. The following step was to extract the altitudes of the ALOS PALSAR, GMTED2010, SRTM, and Topodata DEMs at the same coordinates of the reference points from the official geodetic network. However, we needed to convert the ellipsoidal altitudes of the ALOS PALSAR DEM to orthometric altitudes (geoid) since the GMTED2010, SRTM, and Topodata DEMs were available with altitudes referenced to the geoid (EGM96). For this conversion process, we used the MAPGEO2015 software [30] developed by the *Instituto Brasileiro de Geografia e Estatística* (IBGE) in collaboration with the *Escola Politécnica da Universidade de São Paulo*.

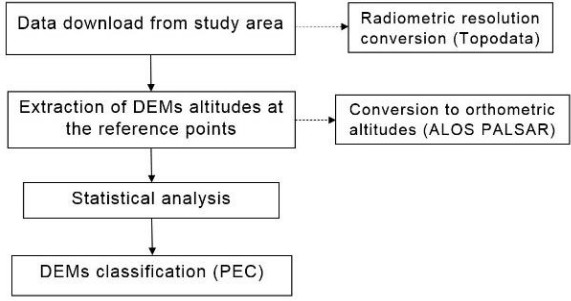

**Figure 3.** Flowchart of methodology.

Afterward, accuracy statistical indicators were calculated such as altimetric error (HE) (1), mean error (ME) (2), mean absolute error (MAE) (3), and root mean square error (RMSE) (4), as performed in some previous studies [9,11,12]. We also analyzed the correlation between the altimetric error and altitude/slope in the study area through the coefficient of determination ($R^2$) (5). Finally, we classified the four DEMs according to the Brazilian Cartographic Accuracy Standard (PEC) [3,31].

$$H_E = H_{REF} - H_{DEM} \tag{1}$$

$$ME = \frac{1}{n} \sum_{i=1}^{n} (H_{REF} - H_{DEM}) \tag{2}$$

$$MAE = \frac{1}{n} \sum_{i=1}^{n} |H_{REF} - H_{DEM}| \tag{3}$$

$$RMSE = \sqrt{\frac{1}{n} \sum_{i=1}^{n} (H_E - ME)^2} \tag{4}$$

$$R^2 = 1 - \frac{\text{RSS}}{\text{TSS}} \tag{5}$$

where $H_E$ = altimetric error; $H_{REF}$ = reference point altitude from Brazilian Geodetic System official altimetric network; $H_{DEM}$ = altitude extracted from DEM at reference point coordinates; ME = mean error; MAE = mean absolute error; RMSE = root mean square error; $n$ = number of reference points; $R^2$ = coefficient of determination; RSS = sum of squares of residuals; and TSS = total sum of squares.

## 3. Results

Results show that regarding the mean error and mean absolute error, the values of the statistical analysis are similar for the four DEMs (Table 3). In fact, we observe that ALOS PALSAR, SRTM, and Topodata DEMs show similarity in all statistical indicators, and it is possible to notice that GMTED2010 shows the worst performance mainly when we consider the RMSE (7.48 m) and the error range (54.00 m), i.e., the difference between the minimum and maximum altimetric errors.

**Table 3.** Statistical metrics of the altitude difference between control points and DEMs.

|  | ALOS PALSAR | GMTED2010 | SRTM | Topodata |
|---|---|---|---|---|
| ME (m) | 12.70 | 13.31 | 12.82 | 12.87 |
| MAE (m) | 12.88 | 13.86 | 12.96 | 13.22 |
| RMSE (m) | 4.95 | 7.48 | 4.76 | 5.38 |
| $H_E$ *min* (m) | −3.58 | −14.22 | −3.21 | −6.17 |
| $H_E$ *max* (m) | 22.04 | 39.78 | 20.93 | 23.60 |
| Error range (m) | 25.62 | 54.00 | 24.14 | 29.77 |

Figure 4 presents the histogram of the altimetric error of each DEM where we can see a positive distortion in all four DEMs and higher variability of the errors in the GMTED2010 product. Nevertheless, we can also notice a very strong correlation between the altitudes of the reference points from the Brazilian official network and the altitudes extracted from the assessed DEMs, where it is possible to verify a determination coefficient ($R^2$) of approximately 0.99 in all of them (Figure 5).

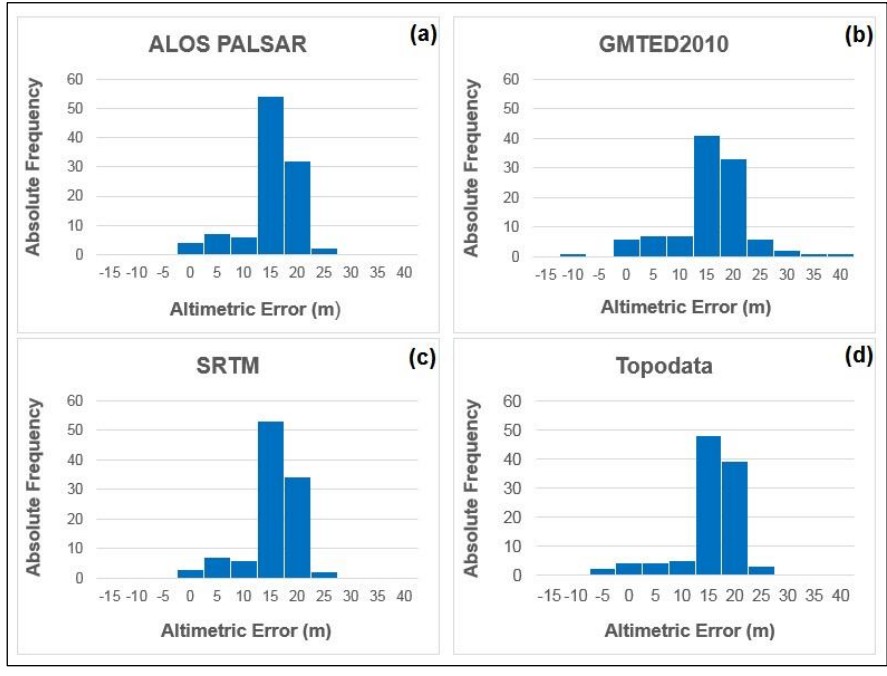

**Figure 4.** Histogram of the altimetric error for ALOS PALSAR (**a**), GMTED2010 (**b**), SRTM (**c**), and Topodata (**d**).

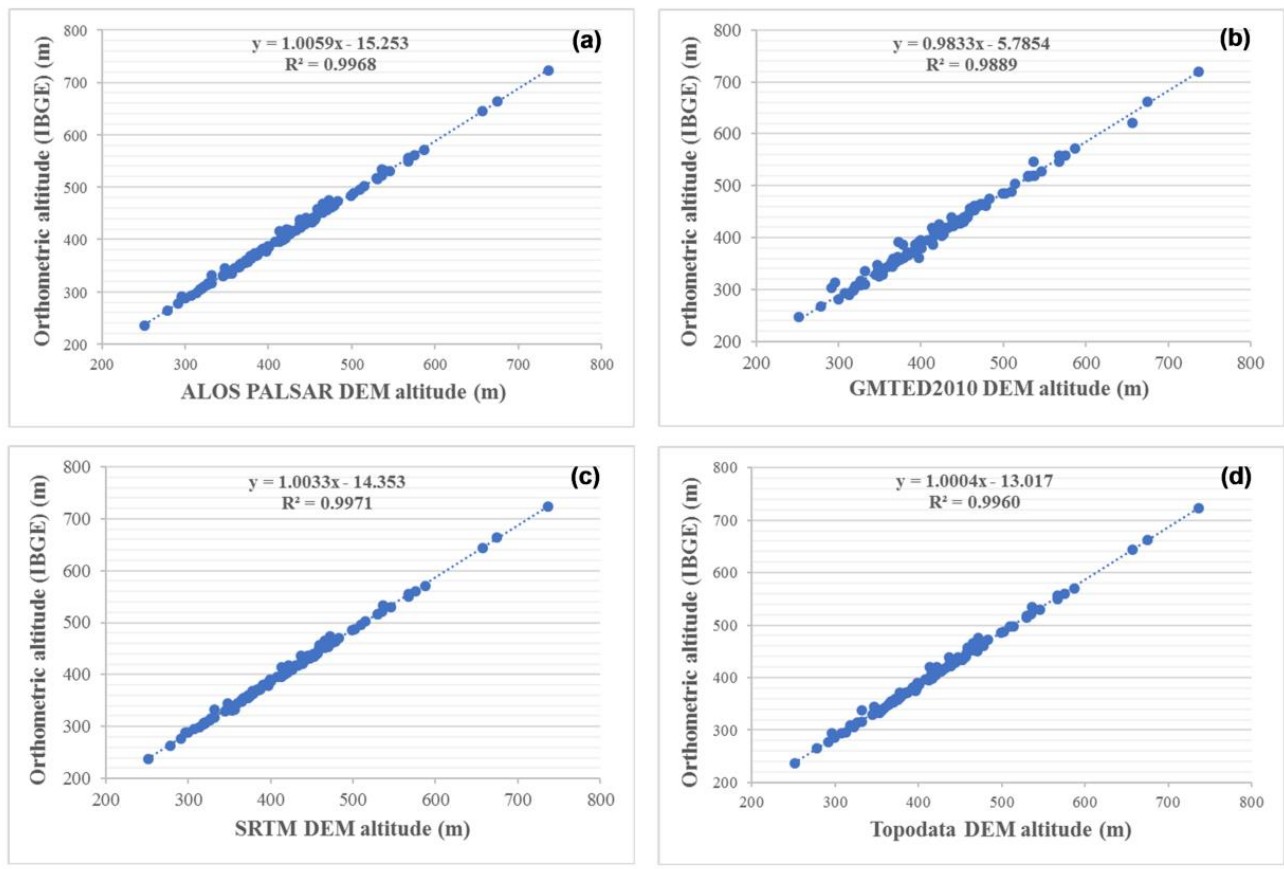

**Figure 5.** Linear correlation between the reference points altitudes of the Brazilian geodetic network and altitudes extracted from each DEM: ALOS PALSAR (**a**), GMTED2010 (**b**), SRTM (**c**), and Topodata (**d**).

Aiming to investigate whether there is a correlation between slope and altimetric error, slope maps of the Balsas River watershed were generated from each DEM, from which six slope classes were established according to IBGE [32] (Figure 6).

The spatial distribution of each slope class in the Balsas River watershed can be seen in Table 4, where we observe that the four DEMs presented approximated values regarding the second slope class (3 to 8%). However, the first class (0 to 3%) shows that the values differ importantly and that SRTM and Topodata presented more similar values in this slope class than the other DEMs. Concerning the other slope classes, ALOS PALSAR, SRTM, and Topodata presented similar results, but the GMTED2010 showed very different results due to its pixel size, which was expected.

**Table 4.** Spatial distribution of each slope class of the Balsas River watershed.

| | ALOS PALSAR | | GMTED2010 | | SRTM | | Topodata | |
|---|---|---|---|---|---|---|---|---|
| Slope | Area (Km$^2$) | % | Area (Km$^2$) | % | Area (Km$^2$) | % | Area (Km$^2$) | % |
| 0 to 3% | 992.55 | 8.04 | 4103.36 | 33.22 | 1776.15 | 14.38 | 2297.57 | 18.60 |
| 3 to 8% | 5459.72 | 44.20 | 5881.54 | 47.61 | 5155.10 | 41.73 | 5295.32 | 42.87 |
| 8 to 20% | 3879.34 | 31.41 | 2075.17 | 16.80 | 3579.51 | 28.98 | 3222.54 | 26.09 |
| 20 to 45% | 1813.29 | 14.68 | 292.36 | 2.37 | 1696.15 | 13.73 | 1454.33 | 11.77 |
| 45 to 75% | 200.78 | 1.63 | 0.07 | 0.00 | 142.65 | 1.15 | 82.15 | 0.67 |
| >75% | 6.83 | 0.06 | 0.00 | 0.00 | 2.94 | 0.02 | 0.59 | 0.00 |
| **Total** | **12,352.50** | **100.00** | **12,352.50** | **100.00** | **12,352.50** | **100.00** | **12,352.50** | **100.00** |

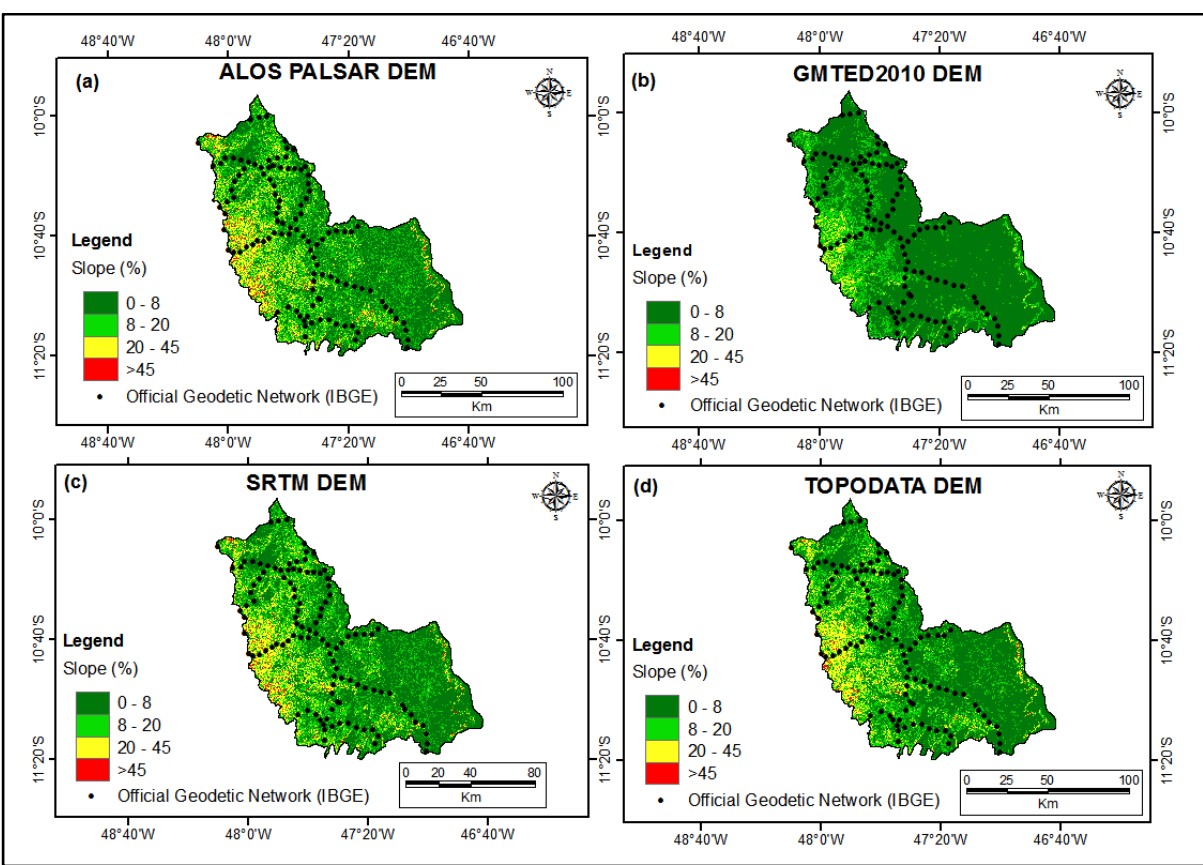

**Figure 6.** Slope map for ALOS PALSAR (**a**), GMTED2010 (**b**), SRTM (**c**), and Topodata (**d**).

In this analysis, no significant linear correlation was observed between slope and altimetric error (Table 5). Nonetheless, it is possible to notice that the RMSE increases as the slope increases in all DEMs except in the ALOS PALSAR DEM.

**Table 5.** Statistical analysis of altimetric error regarding slope classes.

| Slope | ME (m) | MAE (m) | RMSE (m) | $R^2$ | Points |
|---|---|---|---|---|---|
| **ALOS DEM** | | | | | |
| 0 to 3% | 13.89 | 13.89 | 3.81 | 0.0004 | 17 |
| 3 to 8% | 12.94 | 13.21 | 5.03 | 0.0005 | 62 |
| >8% | 11.61 | 11.64 | 4.71 | 0.0088 | 26 |
| | | | | | $\sum$ = 105 |
| **GMTED2010 DEM** | | | | | |
| 0 to 3% | 12.42 | 13.24 | 6.45 | 0.0000 | 54 |
| 3 to 8% | 13.19 | 14.59 | 9.10 | 0.0014 | 43 |
| >8% | 13.45 | 20.96 | 17.57 | 0.0278 | 8 |
| | | | | | $\sum$ = 105 |
| **SRTM DEM** | | | | | |
| 0 to 3% | 14.44 | 14.44 | 2.86 | 0.0005 | 28 |
| 3 to 8% | 12.43 | 12.52 | 4.86 | 0.0175 | 52 |
| >8% | 12.46 | 12.61 | 5.13 | 0.0206 | 25 |
| | | | | | $\sum$ = 105 |

**Table 5.** *Cont.*

| Topodata DEM | | | | |
|---|---|---|---|---|
| **Slope** | **ME (m)** | **MAE (m)** | **RMSE (m)** | $R^2$ | **Points** |
| 0 to 3% | 14.71 | 14.71 | 2.75 | 0.0250 | 38 |
| 3 to 8% | 13.01 | 13.29 | 5.13 | 0.0003 | 47 |
| >8% | 8.88 | 10.01 | 7.06 | 0.0024 | 20 |
| | | | | | $\sum$ = 105 |

We also did not find a significant correlation between altimetric error and altitude, although we noticed a higher value in $R^2$ for all assessed DEMs considering the altitudes above 550 m, except for GMTED2010, as can be seen in Table 6. Regarding ME and MAE, we observe that all DEMs also present the highest values in this same altitude class.

**Table 6.** Statistical analysis of altimetric error regarding altitude.

| ALOS PALSAR DEM | | | | | |
|---|---|---|---|---|---|
| **Altitude (m)** | **ME (m)** | **MAE (m)** | **RMSE (m)** | $R^2$ | **Points** |
| 250–350 | 12.10 | 12.10 | 4.21 | 0.0395 | 20 |
| 350–450 | 13.51 | 13.62 | 4.56 | 0.1384 | 53 |
| 450–550 | 11.52 | 11.99 | 5.92 | 0.0040 | 25 |
| >550 | 13.48 | 13.48 | 2.37 | 0.2223 | 7 |
| | | | | | $\sum$ = 105 |
| **GMTED2010 DEM** | | | | | |
| **Altitude (m)** | **ME (m)** | **MAE (m)** | **RMSE (m)** | $R^2$ | **Points** |
| 250–350 | 11.00 | 14.34 | 11.10 | 0.1037 | 20 |
| 350–450 | 12.92 | 14.40 | 8.79 | 0.0002 | 53 |
| 450–550 | 12.58 | 13.36 | 6.52 | 0.0092 | 25 |
| >550 | 18.01 | 18.01 | 7.73 | 0.0615 | 7 |
| | | | | | $\sum$ = 105 |
| **SRTM DEM** | | | | | |
| **Altitude (m)** | **ME (m)** | **MAE (m)** | **RMSE (m)** | $R^2$ | **Points** |
| 250–350 | 12.00 | 12.00 | 4.00 | 0.0411 | 20 |
| 350–450 | 13.72 | 13.78 | 4.45 | 0.0971 | 53 |
| 450–550 | 11.94 | 12.14 | 5.36 | 0.0002 | 25 |
| >550 | 13.86 | 13.86 | 1.91 | 0.3375 | 7 |
| | | | | | $\sum$ = 105 |
| **Topodata DEM** | | | | | |
| **Altitude (m)** | **ME (m)** | **MAE (m)** | **RMSE (m)** | $R^2$ | **Points** |
| 250–350 | 11.45 | 12.03 | 5.46 | 0.0345 | 20 |
| 350–450 | 13.47 | 13.79 | 5.00 | 0.0870 | 53 |
| 450–550 | 12.18 | 12.46 | 6.06 | 0.0034 | 25 |
| >550 | 14.44 | 14.44 | 2.50 | 0.2060 | 7 |
| | | | | | $\sum$ = 105 |

The interpolated surface of the altimetric error (Figure 7) reinforces that the altimetric error is not related to slope or altitude when we compare it with Figures 2 and 6. In fact, Figure 7 shows very similar surfaces for the SRTM and Topodata DEMs and allows us to verify that the highest altimetric errors coincide with the coordinates of the samples from the Brazilian official network in the central area and that negative errors are concentrated in the southwest region of the watershed.

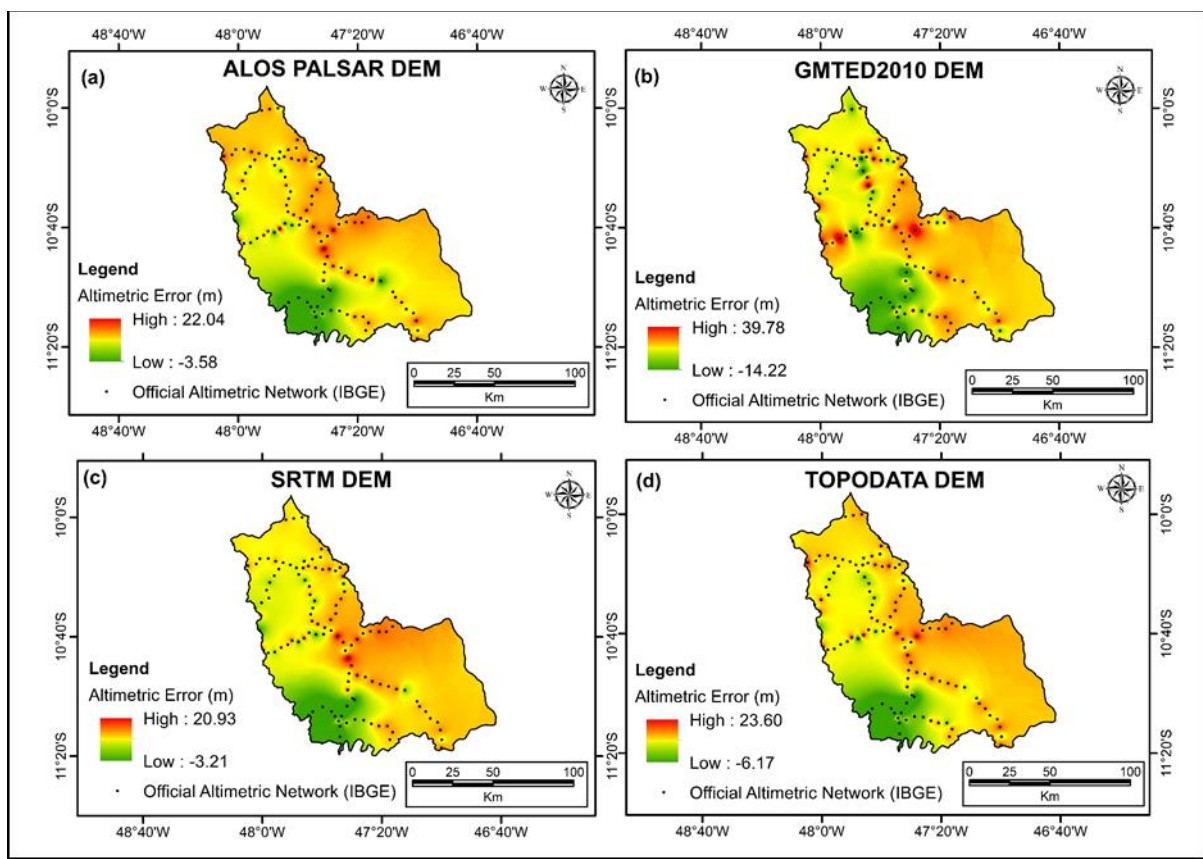

**Figure 7.** Spatial distribution of the altimetric error for ALOS PALSAR (**a**), GMTED2010 (**b**), SRTM (**c**), and Topodata (**d**) elaborated through the inverse distance weighting (IDW) method.

To classify each DEM product according to the appropriate application scale, we used the altimetric cartographic accuracy standard for digital cartographic products production (Table 7), which determines that 90% of point errors collected in the cartographic product must present the same values or less than those predicted in each class.

**Table 7.** Altimetric Cartographic Accuracy Standard of the Elevation Points and the Digital Terrain Model, Digital Elevation Model and Digital Surface Model for Digital Cartographic Products production [7].

| SCALE | 1:25,000 | | 1:50,000 | | 1:100,000 | | 1:250,000 | |
|---|---|---|---|---|---|---|---|---|
| PEC Class | PEC * (m) | RMSE (m) | PEC * (m) | RMSE (m) | PEC * (m) | RMSE (m) | PEC * (m) | RMSE (m) |
| A | 2.70 | 1.67 | 5.50 | 3.33 | 13.70 | 8.33 | 27.00 | 16.67 |
| B | 5.00 | 3.33 | 10.00 | 6.66 | 25.00 | 16.66 | 50.00 | 33.33 |
| C | 6.00 | 4.00 | 12.00 | 8.00 | 30.00 | 20.00 | 60.00 | 40.00 |
| D | 7.50 | 5.00 | 15.00 | 10.00 | 37.50 | 25.00 | 75.00 | 50.00 |

* 90% of point errors collected in the cartographic product must have the same values or less than predicted when compared with the ones surveyed in the field by a high precision method.

Analyzing Table 8, we can verify that the four assessed DEMs can be included in Class B for the 1:100,000 scale and in Class A for the 1:250,000 scale (Table 9) because more than 90% of the extracted points from them had altimetric errors of less than 25 m when compared with the reference points from the Brazilian geodetic network. In addition, the four DEMs also presented an RMSE of less than 16.66 m, as predicted in Table 7.

**Table 8.** Extracted points from the DEMs that had altimetric errors less than 15 and 25 m.

| DEM | $H_E < 15$ m | | $H_E < 25$ m | | |
| --- | --- | --- | --- | --- | --- |
| | Points | % | Points | % | RMSE (m) |
| ALOS PALSAR | 71 | 67.6 | 105 | 100 | 4.95 |
| SRTM | 69 | 65.7 | 105 | 100 | 4.76 |
| Topodata | 63 | 60.0 | 105 | 100 | 5.38 |
| GMTED2010 | 62 | 59.0 | 101 | 96.2 | 6.54 |

**Table 9.** DEMs classification according to Altimetric Cartographic Accuracy Standard for Digital Cartographic Products.

| Scale | ALOS PALSAR | GMTED2010 | SRTM | Topodata |
| --- | --- | --- | --- | --- |
| 1:100,000 | B | B | B | B |
| 1:250,000 | A | A | A | A |

## 4. Discussion

We assessed the vertical accuracy of the ALOS PALSAR, GMTED2010, SRTM, and Topodata DEMs and could classify them according to the Brazilian Cartographic Accuracy Standard. Our results showed that more than 90% of the extracted points from the four DEMs presented altimetric errors less than 25 m when compared with the reference points from the Brazilian geodetic network. Indeed, ALOS PALSAR, SRTM, and Topodata DEMs presented 100% of altimetric errors less than 25 m, and only GMTED2010 DEM presented 3.8% of altimetric errors higher than 25 m. Therefore, the four analyzed DEMs demonstrated satisfactory accuracy to provide mappings in scales up to 1:100,000.

Regarding the statistical indicators, we observed that ALOS PALSAR, and SRTM demonstrated the best performance since ALOS PALSAR had the lowest ME and MAE, while the SRTM showed the lowest RMSE and the smallest error range. The Topodata product presented slightly larger errors when compared with these two DEMs, which can be interpreted as a satisfactory performance since this is a refinement of the SRTM data at 3 arc-seconds (90 m). On the other hand, the GMTED2010 demonstrated the worst accuracy, probably due to its pixel size (231 m), even though it also could be classified in the same accuracy category according to the Brazilian PEC.

According to some studies [33,34], ALOS PALSAR demonstrated a better performance when compared with SRTM and the Advanced Spaceborne Thermal Emission and Reflection Radiometer (ASTER), but in others, when some specific parameters were compared, the SRTM performance was better than ALOS PALSAR [35], ASTER, and GMTED2010 [24,26]. Nonetheless, the Topodata product demonstrated better accuracy in the characterization of drainage networks and watershed vectors when compared with SRTM and ASTER [36].

Although our results have indicated compatibility of the four assessed DEMs with a scale of 1:100,000 regarding the Brazilian Cartographic Accuracy Standard, Moura et al. [3] stated that Topodata, SRTM, and ASTER are compatible with the scale of 1:50,000 in watersheds with little rugged relief. However, in watersheds with higher slopes and higher drainage density, their results also showed compatibility with scales up to 1:100,000 [3].

The above-mentioned findings may indicate that some terrain physical characteristics might influence the results of the DEMs accuracy assessment. Although some studies have found a strong correlation between slope and altimetric error [11,37,38], no significant correlation was observed between these variables in this analysis.

## 5. Conclusions

The acquisition of three-dimensional data from the Earth's surface in the field is a process that requires appropriate equipment and qualified professionals. Furthermore, this process can be expensive and time-consuming, depending on the type of methodology used. In this sense, using DEMs is an attractive alternative for many researchers; consequently, it

is very important to assess their accuracy to ensure their correct applicability concerning the appropriate use scale. Nevertheless, a limitation for assessing the accuracy of DEMs is the absence of accurate data freely available, making fieldwork essential, which makes the assessment process difficult and expensive.

Even though some authors have stated the absence of specific standardized guidelines for DEMs accuracy assessment, in Brazil, the Cartographic Accuracy Standard regulates the quality of cartographic products, and according to this regulation, the four assessed DEMs in this study can supply mappings in scales up to 1:100,000. Regarding the altimetric error, we could not find a significant correlation with altitude or slope, although some authors have found such a correlation in other studies.

A limitation found in this study is that there were few control points from the Brazilian geodetic network inside the Balsas River watershed, and they were badly distributed in the study area because they were located on the banks of the highways. However, the availability of these free data makes possible DEMs accuracy assessment through an accurate data analysis without the need for fieldwork. We suggest that future similar studies be based on the accuracy of a specific application as well in the investigation about other factors that may influence altimetric error, such as watershed roughness, vegetal coverage, and/or land use.

**Author Contributions:** Conceptualization, Zuleide Alves Ferreira and Pedro Cabral; methodology, Zuleide Alves Ferreira and Pedro Cabral; formal analysis, Zuleide Alves Ferreira; investigation, Zuleide Alves Ferreira; writing—original draft preparation, Zuleide Alves Ferreira; writing—review and editing, Zuleide Alves Ferreira and Pedro Cabral; supervision, Zuleide Alves Ferreira and Pedro Cabral. All authors have read and agreed to the published version of the manuscript.

**Funding:** This study was supported by national funds through FCT (*Fundação para a Ciência e a Tecnologia*) under the project UIDB/04152/2020 (*Centro de Investigação em Gestão de Informação* (MagIC) and the *Instituto Federal de Educação, Ciência e Tecnologia do Tocantins* (*Campus Palmas*)).

**Institutional Review Board Statement:** Not applicable.

**Informed Consent Statement:** Not applicable.

**Data Availability Statement:** Not applicable.

**Conflicts of Interest:** The authors declare no conflict of interest.

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
