# Peer review of "A Comparative Study about Vertical Accuracy of Four Freely Available Digital Elevation Models: A Case Study in the Balsas River Watershed, Brazilâ€"

_ijgi, doi:10.3390/ijgi11020106_

Round 1
Reviewer 1 Report
Dear Authors,
Please see in the attachment.

Author Response
Dear reviewer,
Please see the attachment.
Best Regards,
Zuleide Ferreira

Reviewer 2 Report
There are a couple of open questions about this work that should be addressed, or at least thoroughly discussed, by the authors in revision. First, although the authors found "no significant correlation" between slope and altimetric error, though this has been seen in other studies, it is possible that the dataset used here does not provide adequate support to assess this potential correlation. The altimetry data used for validation were collected along roads. Roads are typically not built in areas of steep slopes for safety concerns. From Figure 6, it appears that only one segment of the road-altimetry network may travel through an area with steep slopes. More generally, the second question pertains to whether this study is representative of global DEM accuracy, as it is focused on a small part of Brazil. If the ground-based altimetry data are available throughout Brazil, could the study benefit from considering a large spatial extent with potentially a greater range of elevation (800 m is low for some regions of the world) and greater slopes?
Line 12: "Correct use scale"??? What does this mean? Is the study about accuracy of the product or potential mis-use by end users?
Line 79 and Figure 2: I don't see the value in present all four DEM maps as they look identical; indeed, the results of this study would suggest the DEMs are in such agreement that it's not possible to see a difference between them in print resolution. Perhaps the authors could choose a reference DEM and display difference maps instead? Or, a table of the summary statistics for each map would be sufficient.
Line 176: The authors write: "we can see a positive distortion in the four DEMs and higher variability of the errors in the GMTED2010 product." I think it would be more clear if the authors wrote instead: "...positive distortion in all four DEMs..." This would clarify the positive distortion (bias) is seen all four DEMs whereas the "higher variability of the errors" is seen in a single DEM.
Line 195, Figure 6: Again, difference maps may be more interesting here. Alternatively, a different colorbar stretch may help to emphasize the differences--it looks is if there are extremely few pixels with a slope greater than 45% (and less than 0.1% of pixels are in the highest bin), so perhaps the top two color bins could be re-assigned and those values higher than 45% be clipped at the next-highest bin. A quantile stretch (e.g., bottom 20% of the data in the first bin, next 20% in the next bin) might look better.
*Minor typographical/ grammatical issues:*
Line 29: "DEMs products" should probably be "DEM products"
Line 32: "DEMs products" should probably be "DEM products"
Line 39: "these products creation and availability" should probably be: "the creation and availability of these products"
Line 62: "implanted" should probably be "implemented"
Line 142: "Tododata" should be "Topodata"
Line 280: "However, despite some studies have found..." should instead be written: "Although some studies have found..."
Author Response

(The authors gave the same response as above.)

Reviewer 3 Report
Dear authors,
I reviewed the paper entitled “A comparative study about vertical accuracy of four freely available digital elevation models” by Zuleide Ferreira and Pedro Cabral. The paper analyzes the accuracy of four DEMs, ALOS, SRTM, GMTED2010, and TOPODATA over an area of Brazil, the Balsas River watershed, comparing their vertical heights with those obtained from the Brazilian geodetic system official altimetric network. The paper is well written but some aspects should be clarified.
The title is generic and the study area should be added.
In the abstract, it should be indicated the name of the four DEMs.
It seems that American English is the used style of the English, so check this style along the paper. For example, “levelling” is in British English.
Replace units with symbols.
¿Why has this area been selected and no other with more geodetic information? It should be indicated.
In Figure 2, replace Altimetric Amplitude with Altimetric Range. Although the range is similar, all the DEMs should use the exact legend with the same minimum and maximum value. In such a way, the visual comparison is possible. So use, a scale from 175 to 830 m for example in all the plots.
In the Data section, add a description of the validation data, that is, the Brazilian vertical network used in the study. Name of the leveling lines, number of benchmarks, uncertainties in the heights…
The GMTED2010 DEM has a coarse resolution of 230 m. Considering it, I am wondering if it has sense to use this DEM in the study. You should justify it.
The Methods section should include how the comparison with the official network was done. Every DEM has a different resolution, so, how did you match one cell in the DEM with a certain benchmark. This should be explained. In addition, the methodology section should explain how you assessed the results of your accuracy. Also, how you investigated the correlation between the altimetric errors and slope and altitude.
Table 3. Add ME and MAE when corresponding in the first column. Also replace Amplitude with Range.
Author Response

(The authors gave the same response as above.)

Round 2
Reviewer 1 Report
Dear Authors,
The article meets the minimum requirements for publication.
Regards.
Reviewer 3 Report
I am satisfied with the changes performed in the manuscript.
This manuscript is a resubmission of an earlier submission. The following is a list of the peer review reports and author responses from that submission.